# Towards Greener and Sustainable Airside Operations: A Deep Reinforcement Learning Approach to Pushback Rate Control for Mixed-Mode Runways

## Abstract

Airside taxi delays have adverse consequences for airports and airlines globally, leading to airside congestion, increased Air Traffic Controller/Pilot workloads, missed passenger connections, and adverse environmental impact due to excessive fuel consumption. Effectively addressing taxi delays necessitates the synchronization of stochastic and uncertain airside operations, encompassing aircraft pushbacks, taxiway movements, and runway take-offs. With the implementation of mixed-mode runway operations (arrivals-departures on the same runway) to accommodate projected traffic growth, complexity of airside operations is expected to increase significantly. To manage airside congestion under increased traffic demand, development of efficient pushback control, also known as Departure Metering (DM), policies is a challenging problem. DM is an airside congestion management procedure that controls departure pushback timings, aiming to reduce taxi delays by transferring taxiway waiting times to gates. Under mixed-mode runway operations, however, DM must additionally maintain sufficient runway pressure—departure queues near runway for take-offs—to utilize available departure slots within incoming arrival aircraft steams. While a high pushback rate may result in extended departure queues, leading to increased taxi-out delays, a low pushback rate can result in empty slots between incoming arrival streams, leading to reduced runway throughput.

This study introduces a Deep Reinforcement Learning (DRL) based DM approach for mixed-mode runway operations. We cast the DM problem in a markov decision process framework and use Singapore Changi Airport surface movement data to simulate airside operations and evaluate different DM policies. Predictive airside hotspots are identified using a spatial-temporal event graph, serving as the observation to the DRL agent. Our DRL based DM approach utilizes pushback rate as agent's action and reward shaping to dynamically regulate pushback rates for improved runway utilization and taxi delay management under uncertainties. Benchmarking the learnt DRL based DM policy against other baselines demonstrates the superior performance of our method, especially in high traffic density scenarios. Results, on a typical day of operations at Singapore Changi Airport, demonstrate that DRL based DM can reduce peak taxi times (1-3 minutes, on average); save approximately 27% in fuel consumption and overall better manage the airside traffic.

## 1 Introduction

Airport airside is complex system, characterised by non-linear and non-hierarchical interactions between humans (air traffic controller (ATCO), pilot), machines (aircraft, navigation aids), procedures (safe separation, delay management), and environment (runway, taxiway, gate). Optimal airside operations necessitate the effective utilization of gate, taxiway, and runway resources, achieved through the intricate synchronization of various stochastic and uncertain procedures such as aircraft pushbacks, taxiway movements, and runway take-offs. Inefficient utilization of airside resources may

lead to congestion, delays and reduced runway throughput. Taxi delays, in the US alone, cost approximately $900 million each year in additional fuel burn Chen & Solak (2020). For passengers, delays may cause poor travel experience and missed connections Ali et al. (2019b;a). Increasing carbon emissions resulting from excessive fuel burn at congested airports worldwide have exacerbated environmental concerns as well. Moreover, growing air traffic demand ICAO (2021) will increase aircraft movements on the airside. The consequent increase in airside complexity is expected to cause a non-linear increase in delays Odoni & De Neufville (2003).

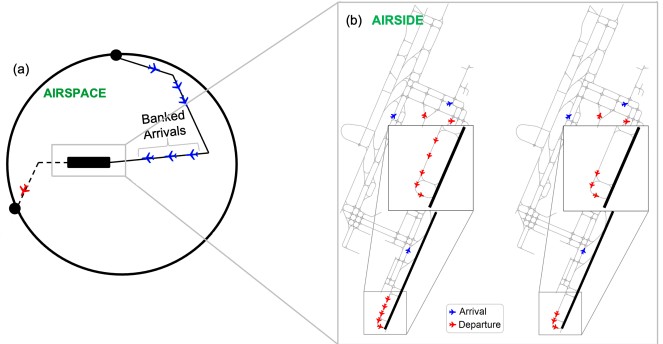 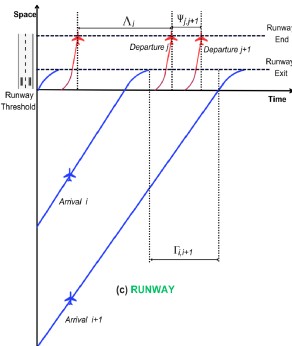

Figure 1: (a) AIRSPACE: Banked arrivals lining up for landing on the mixed-mode runway that is simultaneously servicing departures as well. (b) AIRSIDE: The inset image demonstrates departure metering potential to mitigate surface congestion at Singapore Changi Airport (red and blue icons depict departure and arrival aircraft respectively). Left: Before departure metering, 5 aircraft in queues; Right: After departure metering, 2 aircraft being held and 3 in queues. (c) RUNWAY: A time-space diagram illustrates departures slotted between arriving aircraft. Each aircraft occupies the runway for its designated runway occupancy time, after which the following arrival or departure in the queue can utilize the runway for landing or takeoff.

Mixed-mode runway operations (refer Figure 1(a)) are becoming progressively more prevalent as a strategy to effectively manage the escalating volume of aircraft movements Limin et al. (2022). Notably, Heathrow Airport in the UK, since November 2020, transitioned to mixed mode operations Heathrow (2020). Singapore Changi Airport, in Asia, also plans to increase mixed-mode runway operations by the mid-2020s MITRE (2018). Unlike segregated mode, where each runway is dedicated exclusively to either departures or arrivals, mixed-mode runways can accommodate both departures and arrivals simultaneously ICAO (2004). However, coordinating simultaneous departures and arrivals on the same runway under uncertainties is a challenging task and, owing to the connected nature of airside operations, requires effective co-ordination with aircraft pushbacks from the gate. Referring to the time-space diagram in Figure 1(c), we define the inter departure time between $Departure_j$ and $Departure_{j+1}$ as $\Psi_{j,j+1}$ when there is no arrival in between. The time interval between two consecutive departures with $Arrival_i$ in between is denoted as $\Lambda_i$. Maximizing runway throughput translates to maximally utilizing the inter arrival time $\Gamma_{i,i+1}$ for take-offs, and this involves minimizing both $\Psi_{j,j+1}$ and $\Lambda_i$ Limin et al. (2022). A high pushback rate may result in extended departure queues, leading to increased taxi-out delays and a low pushback rate can result in empty slots between incoming arrival streams, leading to reduced runway throughput. To efficiently utilize the departure slots within the arrival stream, a certain runway pressure, i.e., the number of aircraft in the departure queue waiting for runway availability, should be maintained without causing excessive taxi-out delays. Thus, there is a need to develop intelligent Departure Metering (DM) approaches for mixed-mode runway operations (refer Figure 1(b)).

DM is an airside surface management procedure that involves strategically holding departing aircraft at gates and releasing at appropriate times to mitigate airside congestion. DM assists the departing aircraft to reach the runway just-in-time for take-off while preventing the formation of extensive queues on the airside, as depicted in Figure 1(b). The key idea behind DM is to shift waiting time away from the taxiway, where aircraft engines are running and burning fuel, to the gates where the aircraft can operate in auxiliary power mode Feron et al. (1997). Due to DM, aircraft queuing time can be reduced which in turn may lower fuel consumption, related emissions and improve the airside traffic circulation by having a smoother airside traffic circulation. DM is a sequential

decision-making problem where the decision to release a departing aircraft from the gate not only impacts localised congestion around terminal but also affects movement of arrivals and queuing of departures near runways at a later point in time Ali et al. (2021). Recently, Ali et al. (2022) introduced a DRL based DM policy for segregated mode runway operations. The authors characterized optimal pushback policies as a function of the state of the system. At each time step, the state of the airside traffic was mapped to an on-off action (to meter departures at each terminal) with each state transition generating a reward signal, which was fed back to the DLR agent. The authors reported superior performance of the DRL policy under different traffic densities. Our work introduces novel state and action representations to the DRL framework in Ali et al. (2022) to solve a considerably more challenging mixed-mode runway problem. We perform benchmarking studies with Ali et al. (2022) and another simulation-based optimization method Mori (2017) to demonstrate suitability of our learning-based approach in mitigating airside congestion. Key contributions of this work are as follows:

1. We pose the DM problem of assigning pushback times for mixed-mode runway operations in a DRL based framework. The proposed framework allows for aircraft pushback control by a centralised DM agent. It is noteworthy that we adopt a rate control action instead of an on-off action, because of practical feasibility: After talking to professionally rated ATCOs in Singapore, it's clear they prefer a recommended pushback rate over uniform time intervals instead of an on-off approach, which would require constant intervention.

2. We propose an event graph formulation to encode a variable number of aircraft on the airside into a fixed length vector. This enables an efficient and scalable state representation of the evolving traffic through dynamic airside hotspots to the DRL agent. This method effectively captures airside congestion and builds upon key ideas and prior research in hotspot identification and traffic state representation.

3. We conduct extensive benchmarking experiments to demonstrate the suitability of our method. Our DRL based DM policy is able to predict availability of runway slots for departures amidst the influx of arriving aircraft. It then intelligently schedules pushbacks at optimal times to maximizing runway utilization while minimizing taxi delays. This leads to significant savings in fuel burn without any reduction in runway throughput.

## 2 REPRESENTING AIRSIDE TRAFFIC STATE

The traffic state observation features must be carefully designed to sufficiently distinguish less congested airside traffic states from the more congested ones. Airside congestion is primarily fueled by (a) an imbalance between demand and capacity, coupled with (b) conflicting interactions among aircraft. Spatially, congestion is manifested around taxiway intersections and runway entry/exit where conflictual aircraft interactions occur on the airside. Due to the constraints of limited taxi routes and the requirement to avoid loss of separation, aircraft conflicts are resolved by halting one of the aircraft to yield right of way to the other. This resolution process, while ensuring safety, results in taxi delays for the halted aircraft. Consequently, aircraft conflicts on the airside increase waiting times on the taxiways. Temporally, congestion often occurs during high traffic demand (departure and arrival peaks). When the demand for departures exceeds the runway capacity to handle aircraft movements, congestion ensues. During congestion, departing aircraft may have to follow long queues on taxiways before takeoff, thereby causing taxi-out delays.

To effectively represent state of airside traffic, spatial-temporal movements of aircraft on the airside network are encoded by constructing an airisde event graph. The event graph in this work is inspired from temporal constraint networks Dechter et al. (1991); Kecman & Goverde (2014) to assess time differences between events. The constructed event graph is used to extract observation features for effectively representing airside traffic state to the DRL agent. Airside traffic expected to traverse through hotspots (in a given look-ahead time) can help to predict airside congestion reasonably Ali et al. (2020). Hotspots are clusters of intersections where multiple aircraft may come in close vicinity on the airside Ali et al. (2020). Hotspots serve as reliable indicators of airside congestion; the higher the traffic passing through these areas, the more pronounced the impedance to overall traffic flow becomes, resulting from interactions between aircraft taxiing in conflicting directions. In other words, hotspots contain intersections which facilitate conflicting interactions between aircraft taxiing through them. This method enables the representation of airside congestion in a manner

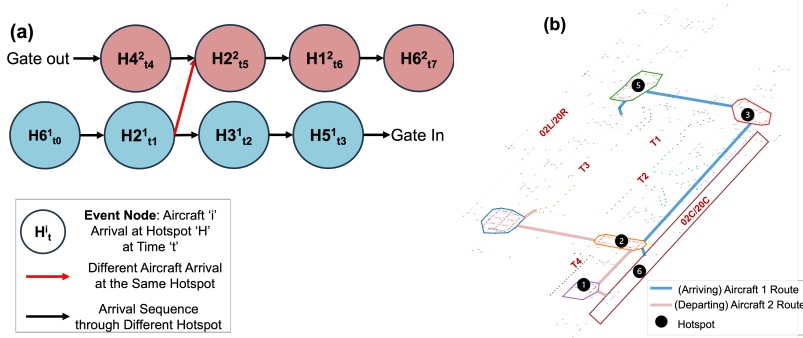

Figure 2: (a) A simplified (showing hotspot events) temporal constraint network representing airside events with precedence relations. (b) Singapore Changi Airport airisde network with blue and pink edges representing taxi routes of an arriving and a departing aircraft respectively. 6 Hotspots- polygons (areas) containing intersection clusters and a mixed mode runway.

that can robustly generalize to differing aircraft count in various traffic scenarios. The following methodology extends the research on hotspot identification Ali et al. (2020) and airside traffic state representation Ali et al. (2022) by constructing a timed event graph that incorporates predictions of aircraft trajectory, through a sequence of hotspots, in the state representation.

**Airside Event-graph and Hotspot Features:** The event-graph, in this study, is a directed acyclic graph $\mathcal{G}(N, E)$. The nodes $N$ encode time-critical events related to aircraft movements, such as gate-out (after pushback), gate-in (after taxi-in), intersection arrival (while taxiing) (refer Figure 2). Assigning timestamps to these events provides chronological context, which is crucial for understanding temporal relationships and interactions between aircraft. A node $i \in N$ is described by ($A_i$, $H_i$, $t_i^{pred}$) representing the unique aircraft id, the id of the taxiway segment or hotspot which the aircraft is expected to enter at time $t_i^{pred}$. Other key information associated with each node is the flight type (arrival or departure), $T_i$, current aircraft velocity, $V_i$, and $t_i^{sch}$, scheduled event time. The directed edges $E$ establish the temporal relationships between events and carry weight ($w_{i,j}$), representing the time intervals between events, effectively capturing the time taken for aircraft to taxi between connected events. An edge models expected aircraft taxiing times and precedence relations between aircraft events. Directed edges are also used to model interactions between aircraft, namely, minimum separation and connection constraints. Edge $(i, j) \in E$ is described by $(i, j, w_{i,j})$ representing the tail event, the head event, and the arc weight. The graph can be constructed based on the scheduled flight plan; each aircraft is described by the taxi route and scheduled taxi start times $t_i^{sch}$. Each edge in the constructed graph should satisfy the following set of constraints.

$$t_j^{pred} \geq t_i^{pred} + w_{i,j} \tag{1}$$

$$t_i^{pred} \geq t_i^{sch}, i \in \{V | type_i = `departure'\} \tag{2}$$

Constraint 1 defines the precedence relation between the tail and the head event of an arc. Constraint 2 represents the pushback constraints for all departure events i.e target start up approval time (actual pushback time after metering) should be greater than scheduled pushback time. The graph topology is built and updated based on the flight schedule and actual positions of aircraft on the network. The model is instantaneously updated when new information becomes available on aircraft positions. After each aircraft position update, a real time trajectory prediction algorithm Tran et al. (2020) derives time estimates of all other events in the graph based on aircraft velocities and assigned taxi routes. This approach explicitly models aircraft dynamics after route conflicts and exploits the available information about the taxiing aircraft. During every graph update, a prediction of aircraft arrival times at each intersection along its taxi route is performed by traversing the graph model in topological order. These predicted values are used to update graph nodes and edges. The graph topology is continuously updated according to the rolling prediction horizon and metering decisions. Possible new aircraft, planned to operate within the horizon, are added to the graph with their

planned route. With every update of aircraft positions, which includes instances of aircraft crossing hotspots, pushback, or landing, the graph undergoes a process where nodes representing past events and their associated incoming and outgoing edges are eliminated (and preserved with the recorded event timestamps). This procedure ensures that the graph's dimensions and complexity are computationally tractable. The airisde event graph helps to predict spatial distribution of aircraft in future traffic states by forecasting each aircraft position based on taxiing speed and assigned taxiway route. It is observed that each aircraft, while taxiing towards its assigned gate or runway, visits multiple hotspots in a specific order based on its taxi route. Rather than encoding the position and velocity of each individual aircraft, as proposed in Ali et al. (2022), which poses challenges in state representation due to the fixed observation space requirements of DRL algorithms like PPO, the encoding of traffic flow through hotspots proves to be a more effective approach (confirmed by the experiments later; refer Figure 3). The constructed event graph is then used to extract current and future traffic density in specific hotspot locations (refer Figure 2 (b)) for effective state representation.

## 3 MARKOV DECISION PROCESS FRAMEWORK

Implementing DRL involves formulating the DM problem as an MDP, producing the main variables (State, Action, Reward, Next State) for each time step: $(s, a, r, s')$. We explain the state-action space and reward structure used to train the DRL based DM agent in the following subsections.

**State Space:** The state observation features capture two crucial types of information: traffic density in hotspot regions and anticipated (new) traffic demand. While hotspots convey congestion information, traffic demand expresses the expected runway pressure. Traffic density in hotspots is extracted following the method explained in previous section. A total of six regions (illustrated as polygons in Figure 2 (b)), containing multiple intersections, are identified as potential hotspots. We encode both present and predicted hotspot traffic density for the upcoming time interval (segmented into three periods of two minutes each) across the six hotspots to represent traffic evolution. The second key information about scheduled traffic demand i.e. count of arrivals and departures, expected to be serviced at the airport in the upcoming time interval (segmented into three periods of two minutes each) is also added to the above hotspot information. Thus, a twenty four dimensional tensor (6X3 + 2X3 = 24) is fed as state observation to the DRL agent.

**Action Space:** In this work, action space refers to the pushback rate i.e number of aircraft to release over the next time steps Simaiakis et al. (2014). Such an action space regulates the rate at which aircraft push back from their gates during high traffic demand periods so that the airport does not reach undesirable, highly congested states. We adopt a rate control action instead of an on-off action—as implemented in Ali et al. (2021; 2022), because of practical feasibility: After talking to professionally rated ATCOs in Singapore, it's clear they prefer a recommended pushback rate over time instead of an on-off approach, which would require constant intervention. However, we find that this action space performs better an on-off action (refer Figure 3). In this paper, the DRL agent can select to pushback $\{0, 1, 2, 3, 4\}$ aircraft over the next $n$ time steps. In other words, the agent can choose not to initiate pushback for any aircraft or up to 4 aircraft within a decision-making interval spanning 3 simulation time steps (equivalent to 30 seconds) specifically.

**Reward Structure:** The reward structure is designed to encourage high aircraft taxiing speeds (less delays) and runway utilization (high throughput). Consequently, the reward at each time step is a function of by three key elements: pushback action, taxiway movements, and runway usage.

$$R = R_{Taxiway} + R_{Action} + R_{Runway} \tag{3}$$

The introduction of $R_{Taxiway}$ aims to discourage slow taxi speeds, leading to negative rewards for aircraft starting from rest after pushback. In the absence of $R_{Taxiway}$, the agent prioritizes releasing aircraft immediately to earn rewards quickly (due to the discount factor), causing queues and delays. When the taxiway is occupied, it equals $\alpha_1$; otherwise, $\alpha_2$ (see Table 1). $R_{Taxiway}$ is negative until the aircraft achieves high velocity. Therefore, without $R_{Action}$, the model learns to stop all aircraft at the gate to avoid negative rewards. To promote pushback actions, $\alpha_3$ is awarded, when an aircraft is pushed back for offsetting the negative effects of $R_{Taxiway}$, initially. The $R_{Runway}$ incentivizes the agent to push back more aircraft and use available runway slots effectively. Without $R_{Runway}$, the agent aims to optimize $R_{Action} + R_{Taxiway}$ for releasing aircraft and managing congestion. However, this leads to a DM policy releasing aircraft one by one, causing a significant reduction in

runway throughput compared to non-metering scenarios. This study incorporates $R_{Runway}$ ($\alpha_5 = 0$ when the runway is occupied, $\alpha_6 = -4$ when vacant) in the reward structure. Experimental iterations determine values for $\alpha_1$-$\alpha_6$ (see Table 1).

| Events | Reward |
|---|---|
| Taxiway utilization ($\alpha_1$) | $\sum_k (0.6V_k - 0.5)$, where k is count of departures moving on taxiways |
| Vacant taxiway ($\alpha_2$) | -40 per time step |
| Pushback ($\alpha_3$) | +1 |
| NO pushback ($\alpha_4$) | 0 |
| Occupied runway ($\alpha_5$) | 0 |
| Vacant runway ($\alpha_6$) | -4 per time step |

Table 1: Reward structure to find DM policy.

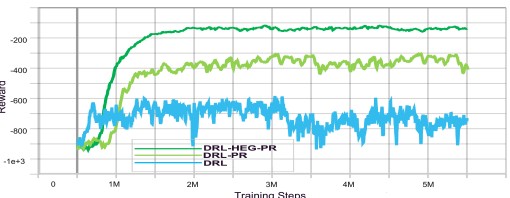

Figure 3: Reward curves of DM policies: DRL, DRL-PR and DRL-HEG-PR.

## 4 EXPERIMENTS

For the purpose of training the DM agent with realistic airside traffic scenarios, Advanced Surface Movement Guidance and Control System (A-SMGCS) data of Singapore Changi Airport is utilized to simulate traffic demands at runway 02C/20C. A-SMGCS dataset contains aircraft movement information over a period of 4 months (Oct-2017 to Jan-2018; 121 days). In this work, we have incorporated four sources of uncertainty in the airside simulator. Aircraft start times (TOBTs, ELDTs) and the intervals between them (inter-departure and inter-arrival times) are simulated based on observed distributions in historical data. Further, uncertainties in taxi routes allocation and aircraft taxi speeds (due to conflictual interactions) have also been incorporated into the simulated scenarios. Moreover, to expose the agent to varying traffic densities, the hourly traffic density is manipulated artificially by either condensing or dispersing it Ali et al. (2022). This approach to generating episodes aids in exploring the impact and appropriateness of the learned policy across different traffic density scenarios within the existing airside taxi network. In each episode, aircraft start times are randomized, and inter-event times are adjusted proportionally to reflect lower or higher demand. This dynamic adjustment prevents the DM policy from overfitting to the historical traffic patterns in data.

Proximal Policy Optimization (PPO) Schulman et al. (2017) is selected to train the DM agent. PPO is used for departure metering by Ali et al. (2022) and found suitable for our event graph-based state space and pushback rate-based action space. To find an optimal DM policy, we have used the implementation of the Stable Baselines implementation Hill et al. (2018) for PPO to train the DM agent. Different experiments for selecting the network architecture and activation function were performed and as a result, $Tanh$ activation function and the separated actor and critic networks are selected (refer to Figure 7 in APPENDIX). Hereafter, DRL-HEG-PR shall refer to our DRL model with selected hyper-parameters. The simulation environment is discrete and we select a 30-second update interval (corresponding to three time steps) as the agent decision-making frequency.

**Policy Convergence During Training:** We preform ablation experiments to investigate the impact of the novel state and action space on the convergence performance of DRL algorithms during training. In Figure 3, we report the reward curves pertaining to the algorithms DRL (baseline algorithm; refer Ali et al. (2022)), DRL-PR (DRL with Pushback Rate as action space) and DRL-HEG-PR (DRL with Hotspot Event Graph as state space and Pushback Rate as action space). Ablation experiments—with and without novel state and action space—show that the state observations based on hotspot-event graph alongwith action space based on departure rate, lead to improved returns and quicker convergence (refer Figure 3). This is likely due to the fact that the evolution of traffic density in and around intersections serves as a reliable indicator of aircraft interactions and airport congestion. Also, pushback rate based action space can better leverage the encoded traffic density information in observation space than multi-binary action space used in DRL Ali et al. (2022) (which is more suitable for terminal level spot metering).

**Experiment Set 1:** These experiments aim to assess the potential advantages of the DM policy in average traffic density (40 aircraft(movements)/hour) operations at Singapore Changi Airport Ali et al. (2020). This traffic density aligns with what is typically observed at medium to large-sized airports, such as Charles de Gaulle (Paris; CDG) Badrinath et al. (2020) and Charlotte Douglas International (North Carolina; CLT) Badrinath et al. (2020). We evaluate taxi-out delays, gate hold

times, and aircraft conflicts with and without the DM policy, using the same unseen scenarios (refer $TOT^{delay}$, $GH$ in APPENDIX) .

**Experiment Set 2:** The 2nd set of experiments intends to benchmark the performance of our DRL-HEG-PR based DM with DRL Ali et al. (2022) and TS Mori (2017). In actual airside operations, the number of aircraft varies throughout the day. For example, although airports like CDG and CLT witness average traffic of about 40 aircraft per hour, CLT experiences higher peak demand for departures than CDG. This leads to times when the demand exceeds the airport's capacity, resulting in longer queues forming Badrinath et al. (2020). The performance of DRL-HEG-PR, DRL and TS are evaluated by running experiments with varying traffic densities. These experiments maintain an equal ratio of arriving and departing flights (1:1) to ensure fair comparisons of taxi-out-delays, gate hold times, and runway throughput, both with and without DM, in the same unseen situations (refer $TOT^{delay}$, $GH$, $R$ in APPENDIX). For each traffic level, the evaluation is conducted with 30 scenarios to ensure thorough and accurate results.

**Experiment Set 3:** The 3rd set of experiments aims to showcase the practical implementation of the DRL-HEG-PR-based DM policy for metering an entire day of operations at Singapore Changi Airport. In these experiments, simulated traffic scenarios for a typical day's traffic density (approximately 1000 aircraft/day) are generated using randomization while adhering to the observed hourly demand patterns from real-world data. This setup ensures that the model's performance is assessed under diverse conditions, encompassing variations in both aircraft fleet size and composition. Furthermore, the benefits of the DM policy in reducing fuel consumption over an entire day of operations are examined. This evaluation involves comparing fuel consumption estimates obtained with and without the application of the DM policy. The fuel consumption estimation is conducted using a well-established methodology Chen et al. (2015b;a) for calculating ground fuel burn using the aviation emission databank from the International Civil Aviation Organization (ICAO).

## 5 RESULTS

### 5.1 EXPERIMENT SET 1: DM RESULTS IN AVERAGE TRAFFIC DENSITY SCENARIOS

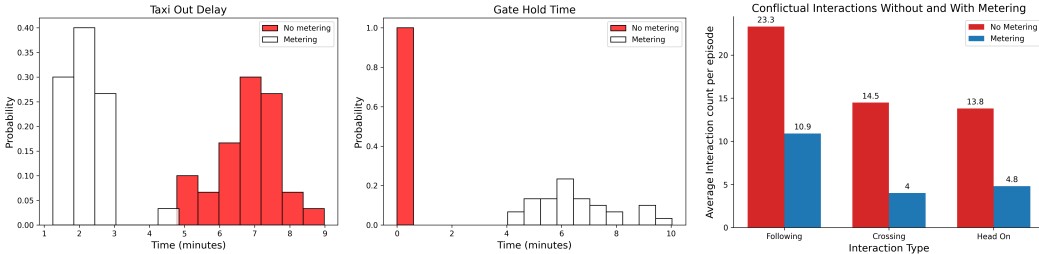

Figure 4: **Left:** Reduction in taxi-out-delay due to DM policy. The delay distribution shifts left towards lower delay values implying lower airside delays and congestion. **Middle:** Gate holds due to DM policy. Taxi delays are transferred to the gates. **Right:** Reduction of spatial conflicts due to metering.

**Taxi-Out-Delay, Gate Hold Time and Spatial Conflicts:** Figure 4 (Left) illustrates a substantial decrease in taxi-out delays attributed to the learned DM policy. The delay distribution shifts leftward towards lower delay values, with the mean delay decreasing from approximately 6.8 ($\pm$ 0.9) to 2.2 ($\pm$ 0.6) minutes, marking an 87% reduction in taxi delays at a traffic density of 40 aircraft per hour. The reduction in taxi delays, however, come at the cost of increased gate hold times. Figure 4 (Middle) demonstrates that due to the DM policy, gate holds of up to 9.6 minutes are incurred in metering scenarios.

Considering the limited taxi routes to destinations and the need to adhere to separation constraints, aircraft may interact and potentially conflict with other aircraft if allowed to move freely. These conflicts can fall into three categories: following, crossings, and head-on conflicts Duggal et al. (2022). All types of conflicts lead to delayed movements on taxiways as they must be resolved by halting one of the aircraft to prevent the risk of compromised separation or collision. Figure 4 (Right)

Table 2: Impact of airside traffic density on gate holds, taxi times, and runway throughput; mean and standard deviation values of the difference in the distribution in minutes (gate holds and taxi times) and take-offs per hour (runway throughput).

| Aircraft Count in Scenario | TS | | | DRL | | | DRL-HEG-PR | | |
|---|---|---|---|---|---|---|---|---|---|
| | Additional Gate Holding | Taxi-Out-Delay Reduction | Runway Throughput Reduction | Additional Gate Holding | Taxi-Out-Delay Reduction | Runway Throughput Reduction | Additional Gate Holding | Taxi-Out-Delay Reduction | Runway Throughput Reduction |
| 10 | $0.1 \pm 0.1$ | $\mathbf{0.1 \pm 0.1}$ | $0.3 \pm 0.0$ | $5.3 \pm 8.5$ | $-0.1 \pm 0.5$ | $0.1 \pm 0.1$ | $0.1 \pm 0.1$ | $0 \pm 0.1$ | $\mathbf{0.0 \pm 0.0}$ |
| 30 | $3.3 \pm 1.6$ | $2.6 \pm 1.1$ | $0.5 \pm 0.0$ | $24.1 \pm 7.3$ | $\mathbf{4.8 \pm 1.3}$ | $0.4 \pm 0.2$ | $6.4 \pm 1.4$ | $4.6 \pm 0.8$ | $\mathbf{0.0 \pm 0.0}$ |
| 50 | $11.4 \pm 2.1$ | $4.3 \pm 2.6$ | $0.6 \pm 0.0$ | $18.6 \pm 6.7$ | $11.3 \pm 3.1$ | $0.2 \pm 0.2$ | $18.3 \pm 1.5$ | $\mathbf{13.7 \pm 1.9}$ | $\mathbf{0.0 \pm 0.0}$ |
| 70 | $22.1 \pm 2.1$ | $3.5 \pm 3.5$ | $0.6 \pm 0.0$ | $7 \pm 5.9$ | $4.8 \pm 5.2$ | $0.0 \pm 0.0$ | $33.3 \pm 2.4$ | $\mathbf{24.2 \pm 3.2}$ | $\mathbf{0.0 \pm 0.0}$ |

illustrates the impact of DM on spatial conflicts, with all types of conflicts decreasing by at least a factor of 2. Consequently, the overall reduction in taxi delays can be partly attributed to the decrease in spatial conflicts resulting from the implementation of DM.

## 5.2 EXPERIMENT SET 2: BENCHMARKING DRL-HEG-PR WITH TS MORI (2017) AND DRL ALI ET AL. (2022) IN DIFFERENT TRAFFIC DENSITY SCENARIOS

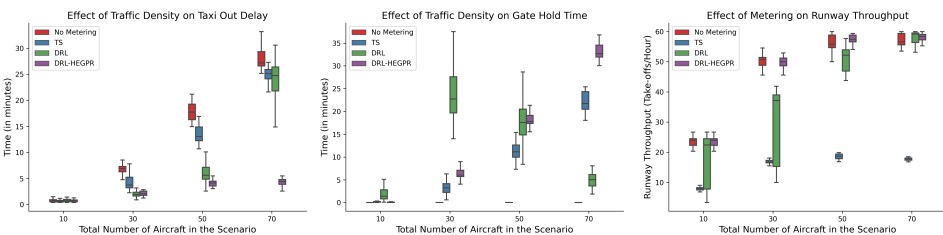

Figure 5: Left: The comparison, between the proposed approach and other baselines, in terms of taxi-out-delay with an increase in traffic density. Middle: The comparison, between the proposed approach (DRL-HEG-PR) and other baselines, in terms of average gate hold with an increase in traffic density. Right: The comparison, between our proposed approach (DRL-HEG-PR) and other baselines, in terms of runway throughput with an increase in traffic density.

**Taxi-out-delay reduction and additional gate holds with increasing traffic density:** Figure 5 illustrates a general trend of increasing taxi-out-delay and gate holds with higher traffic density. Taxi-out-delays show a linear rise under non-metering conditions, but the TS, DRL, and DRL-HEG-PR DM policies effectively mitigate this increase. Furthermore, the difference in these metrics between metered and non-metered scenarios widens as traffic density increases, suggesting that the potential benefits of the DRL-HEG-PR policy are more pronounced in denser traffic.

In high traffic density scenarios, DRL-HEG-PR outperforms TS and DRL (see Table 2). While TS and DRL marginally excel (by 0.1-0.2 minutes) in low traffic scenarios, the significance lies in managing congestion during high traffic. For instance, as airside traffic surpasses 30 aircraft, DRL-HEG-PR achieves 9-21 minutes and 2-20 minutes better taxi-out-delay reduction compared to TS and DRL, respectively (see Table 2). Moreover, with increased traffic density, average gate hold times peak at around 20 minutes for TS and 30 minutes for DRL-HEG-PR. Interestingly, DRL exhibits a less defined pattern, with more pronounced gate holds in scenarios involving 30 aircraft than in those with 70 aircraft. This disparity is likely due to the insufficient encoding of future airside traffic dynamics in the state representation, a crucial element for effective congestion management in mixed mode operations. In summary, DRL stands out for its ability to alleviate taxiway congestion better than TS and DRL, achieving more efficient delay transfers to the gate.

**Runway throughput:** Under non metering, a high runway throughput is anticipated due to the immediate release of all departures, resulting in elevated runway pressure. However, this approach leads to extensive queue formation and increased fuel consumption at the runway. For efficient airside operations, a DM policy needs to learn how to effectively balance taxi delays with runway throughput amid uncertainties. Both TS and DRL experience a decline in throughput of up to 0.6 and 0.4 aircraft movements per hour, respectively, due to suboptimal gate holds (see Figure 5 and TABLE 2). In contrast, DRL-HEG-PR demonstrates ability to predict runway slot availability for

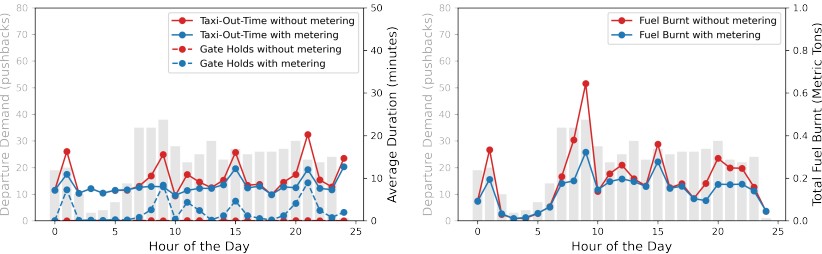

Figure 6: Average taxi-out and gate hold times (left); Fuel burnt (Right) on a typical day at Singapore Changi Airport with and without DM.

departures amidst the incoming flow of arriving aircraft. This policy strategically schedules pushbacks for departing flights at optimal times to fully exploit available slots while minimizing taxi-out delays (see the previous subsection on taxi delay reduction).

## 5.3 Experiment Set 3: DM Results for One Day Traffic Scenarios

Figure 6 illustrates the application of the learnt DM policy to regulate an entire day of operations at Singapore Changi Airport. Notably, a correlation is evident between spikes in departure demand and peaks in taxi times (see Figure 6 (Left)). During periods from 03AM to 06AM, the demand for departure pushbacks is low, resulting in the absence of gate holds. Overall, the DM scenarios effectively manage taxi times, as indicated by lower taxi time peaks, achieved through the transfer of delays from taxiways to gates (see Table 3).

Figure 6 (Right) also illustrates the comparison of fuel consumption at Singapore Changi Airport with and without the intervention of the DM policy. Generally, an increase in departures corresponds to higher fuel consumption. However, since fuel usage is influenced by thrust, which is tied to aircraft acceleration and deceleration Chen et al. (2015b;a), fuel consumption peaks align with frequent stop-and-go movements and conflicting interactions that lead to prolonged taxi times. The learned DM policy

Table 3: Impact of DRL-HEG-PR on taxi-out time and fuel burn in one-day scenarios.

| Time of the Day (Local Time) | Pushback Demand (count) | Reduction in taxi-out Time (min.) | Reduction in Fuel Consumption (Metric Tons) |
|---|---|---|---|
| 12 AM-06 AM | 63 | 1.9 | 0.1 |
| 06 AM-12 PM | 172 | 3.2 | 0.6 |
| 12 PM-06 PM | 156 | 1.4 | 0.2 |
| 06 PM-12 AM | 153 | 2.3 | 0.4 |

effectively reduces potential conflicting interactions, leading to smoother aircraft movement on the airside and subsequently lowering fuel consumption. Consequently, the total fuel consumption over the course of the day decreases by approximately 26.6%, from 4.7 to 3.5 metric tons (refer to Figure 6 and Table 3).

## 6 Conclusion

In this paper, we formulated DM using a DRL-based approach to minimize aircraft taxi delays while maintaining optimal runway throughput in mixed-mode runway operations. The proposed approach employed an actor-critic algorithm, PPO, along with a novel state and action space to learn an effective DM policy in an inherently uncertain airport environment. Extensive experiments demonstrate that DRL-based DM is able to maintain sufficient runway pressure for on-time delivery of departing aircraft in slots between scheduled arrival streams with significantly lesser taxi-out delays than the baseline algorithms, especially under high traffic density scenarios. The learnt DM policy is applied to Singapore Changi Airport scenario as a case study. Results highlight that the DM policy can effectively contain aircraft taxi times even under highly congested situations while reducing fuel burn and environmental impact. Next, we aim to investigate the transfer learning potential of the learnt DM policies across airports.

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

## A    APPENDIX

**Mathematical definition of metrics to measure performance of airside operations:** Taxi-out delay, for an aircraft $i$, is computed by subtracting unimpeded taxi-out-time ($TOT_i^{unimpeded}$) from actual taxi-out-time ($TOT_i^{actual}$) observed in simulation. Average taxi-out delays ($TOT^{delay}$) are then computed by using equation 4.

$$TOT^{delay} = \frac{\sum_i (TOT_i^{actual} - TOT_i^{unimpeded})}{k} \tag{4}$$

where, k refers to count of departures in the scenario. The unimpeded taxi-out-time of an aircraft $i$ depends on its assigned route. It is computed based on taxi path length ($l_i$) and the maximum allowable taxi speed ($v_i^{max}$) of the aircraft using equation 5.

$$TOT_i^{unimpeded} = \frac{l_i}{v_i^{max}} \tag{5}$$

Average taxi-out-time ($TOT$) for a scenario is computed by averaging actual taxi-out-times of all departures using equation 6.

$$TOT = \frac{\sum_i TOT_i^{actual}}{k} \tag{6}$$

Average gate hold time ($GH$) for a scenario is computed by averaging gate hold times of all departures using equation 7.

$$GH = \frac{\sum_i (TSAT_i - TOBT_i)}{k} \tag{7}$$

where $TOBT_i$ and $TSAT_i$ are the scheduled and actual pushback times, of departure $i$, respectively.

Runway throughput ($R$) for a scenario is computed by dividing number of successful take-offs and landings with the runway makespan.

$$R = \frac{k + g}{T} \tag{8}$$

where $T$ is the runway makespan i.e. total time taken to serve $k$ departures and $g$ arrivals.

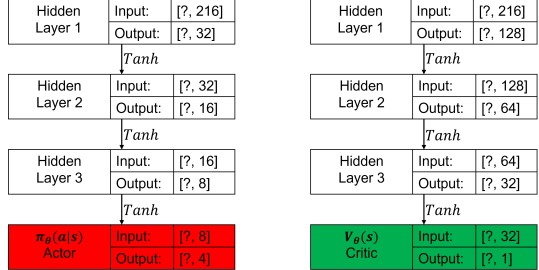

Figure 7: Multilayer Perceptron based neural network architecture used in PPO. In both actor and critic networks, $Tanh$ activation function is used after each linear layer.

