# OpenReview forum: "Towards Greener and Sustainable Airside Operations: A Deep Reinforcement Learning Approach to Pushback Rate Control for Mixed-Mode Runways"
_ICLR.cc/2024/Conference — Submitted to ICLR 2024_

### Official Review · Reviewer_NYw3 · 2023-10-30

**Soundness:** 2 fair
**Presentation:** 3 good
**Contribution:** 2 fair
**Rating:** 3
**Confidence:** 3

**Summary:**

This paper considers using deep reinforcement learning to design pushback rate for mixed-mode runways. Based on the congestion information, a controller decides the pushback rate (from 0 to 4). The reward is a combination of how fast aircraft leave the gate and the time it spends waiting to take off.

**Strengths:**

+ Optimizing aircraft traffic within an airport is a complex and important problem.
+ It's easy to understand what the paper is trying to do and the problem makes sense.
+ Some numerical improvement are seen against an uncontrolled policy.

**Weaknesses:**

- It's not clear that deep reinforcement learning is the right tool to use here. Since the decision is centralized, and there are important safety constraints, a rolling horizon approach (e.g., a MPC) may do better.
- The learning problem setup is also fairly standard and it's hard to see innovations in that regard. As the authors point out, a rate would be more natural for the traffic controllers. Looking at how this can be directly learned rather than taking a discretization approach as currently done in the paper would be interesting.

**Questions:**

- In current practice, is the pushback process entirely unmetered? Or would an aircraft need the clearance from air traffic control to pushback? If the latter is true, the ATC would be implicitly doing some optimization right?
- Does the methods in the paper use the fact that a runway is mixed use? I'm not sure I see that in the algorithm. Also, for a large airport like Changi, are the runways mixed use?
- Sometimes a plane pushes back to free up a gate such that a landed aircraft can use it. So holding planes at gates is not exactly zero cost.

---

> ### Author Response · Authors · 2023-11-22
> **Response to reviewer questions**
>
> Thank you for your comments. Below are our responses to your questions.
>
> **1. In current practice, is the pushback process entirely unmetered? Or would an aircraft need the clearance from air traffic control to pushback? If the latter is true, the ATC would be implicitly doing some optimization right?**
>
> In most medium to large airports, systems like the Advance Surface Movement Guidance Control System (A-SMGCS) recommends pushback schedules and the final pushback clearances are issued by Air Traffic Controllers (ATCOs). Therefore, in our paper, we also benchmarked our method against non metering baseline which is the schedule based on historical surface movements.
>
> **2. Does the methods in the paper use the fact that a runway is mixed use? I'm not sure I see that in the algorithm. Also, for a large airport like Changi, are the runways mixed use?**
>
> Yes, as per the historical A-SMGCS data analysis, the Changi Airport does operate in mix mode. Operating runways in segregated or mix mode depends on traffic demand, weather, wind conditions etc.  For further clarity, we have modified the section 3 to explicitly highlight the mix mode aspects in the state space formulation. The updated sub-section now reads as follows:
>
> ### State Space:
> The state observation features capture two crucial types of information: traffic density in hotspot regions and anticipated traffic (both arrivals and departures) demand. While hotspots convey congestion information, traffic demand expresses the expected mix mode runway pressure. Traffic density in hotspots is extracted following the method explained in previous section. A total of six regions (illustrated as polygons in Figure 2 (b)), containing multiple intersections, are identified as potential hotspots. We encode both present and predicted hotspot traffic density for the upcoming time interval (segmented into three periods of two minutes each) across the six hotspots to represent traffic evolution. The second key information about scheduled traffic demand i.e. count of arrivals and departures, expected to be serviced at the mix mode runway in the upcoming time interval (segmented into three periods of two minutes each) is also added to the above hotspot information. To indicate that a runway is mix mode operated, count of arrivals and departures, instead of being summed up, are represented as distinct tensor elements to the DRL agent. Thus, a twenty four dimensional tensor (6X3 + 2X3 = 24) is fed as state observation to the DRL agent.
>
> **3. Sometimes a plane pushes back to free up a gate such that a landed aircraft can use it. So holding planes at gates is not exactly zero cost.**
>
> Yes, gate conflicts arising due to departure metering has been studied extensively in previous research Ali et al. 2022; Chen and Solak 2020. To resolve gate conflicts with incoming arrivals, departing aircraft can be towed to metering spots or areas on the apron/taxiways where the aircraft wait until the pushback clearance is given by ATCOs. In this study, taxi delays are calculated from the moment an aircraft receives pushback clearance (starts taxiing) until it takes off from the runway.
>
> **References**
>
> Ali, H., Pham, D. T., Alam, S., & Schultz, M. (2022). Integrated airside landside framework to assess passenger missed connections with airport departure metering. In Proc. Int. Conf. Res. Air Transp. (pp. 1-9).
>
> Chen, H., & Solak, S. (2020). Lower cost departures for airlines: Optimal policies under departure metering. Transportation Research Part C: Emerging Technologies, 111, 531-546.

---

### Official Review · Reviewer_WotT · 2023-10-31

**Soundness:** 2 fair
**Presentation:** 2 fair
**Contribution:** 2 fair
**Rating:** 3
**Confidence:** 4

**Summary:**

This research introduces a Deep Reinforcement Learning (DRL) strategy for Departure Metering (DM) during mixed-mode runway operations, focusing on optimizing pushback timings to alleviate airside congestion. By framing the DM challenge as a markov decision process and utilizing Singapore Changi Airport data, the study simulates airside activities to assess DM policies. The DRL agent uses spatial-temporal event graphs to detect airside hotspots and adjusts pushback rates dynamically. Compared to other methods, the DRL approach proves superior, especially under high traffic. Findings indicate notable reductions in taxi times and a 27% fuel savings at Singapore Changi Airport.

**Strengths:**

1. The author employs an "airside event graph", inspired by temporal constraint networks, to effectively represent the state of airside traffic. This novel representation captures spatial-temporal movements of aircraft, offering a sophisticated modeling approach.
2. It considers various types of conflicts (e.g., following, crossings, and head-on conflicts) that can occur during taxiing. This comprehensive approach ensures that the proposed solution addresses a wide range of operational challenges.
3. The paper doesn't just propose a solution, but also quantitatively evaluates its impact, as seen in the results section. This rigorous evaluation approach, including fuel consumption analysis, provides tangible evidence of the proposed method's efficacy.

**Weaknesses:**

1. The model primarily validates using the Singapore Changi Airport scenario, questioning its adaptability. The research falls short in demonstrating generalization across varied environments.
2. The "hotspots" paradigm is ambiguously defined. Such lack of clarity in state representation can lead to convoluted state spaces and suboptimal policies in reinforcement learning.
3. The reward structure is inadequately elaborated. Given its importance in shaping agent behavior in reinforcement learning, its cursory treatment raises concerns about potential biases and pitfalls.
4. The action space, narrowed to pushback rate control, oversimplifies the complexity of airside operations, missing out on capturing nuanced dynamics.
5. The comparative evaluation against baselines lacks depth and rigor, failing to provide a comprehensive assessment against state-of-the-art methods.

**Questions:**

1. How does your approach compare with existing MARL algorithms in terms of efficiency and efficacy? Are there benchmark comparisons against state-of-the-art multi-agent methods to validate the superiority of your approach?
2. How do agents in the model communicate during airside operations?
3. Reward shaping and credit assignment are critical in multi-agent settings. How do you ensure that individual agents receive appropriate credit for their actions to promote cooperative behavior? Are there specific reward shaping techniques employed to foster collaborative actions?
4. How does the method explore the joint action space and how scalable is it with increasing dimensions?

**Details Of Ethics Concerns:**

Looks fine.

---

> ### Author Response · Authors · 2023-11-22
> **Response to reviewer questions**
>
> Thank you for your comments. Below are our responses to your questions.
>
> **1. How does your approach compare with existing MARL algorithms in terms of efficiency and efficacy? Are there benchmark comparisons against state-of-the-art multi-agent methods to validate the superiority of your approach?**
>
> Our approach is designed considering that airport airside ground movements are centrally managed by Air Traffic Controllers (ATCOs). Given this centralized management, we have opted for a centralized agent paradigm to formulate the departure metering problem. Consequently, our approach is not formulated as a multi-agent problem, and thus, direct comparisons with existing MARL algorithms are not applicable.
>
> **2. How do agents in the model communicate during airside operations?**
>
> As our approach doesn't treat the departure metering problem as a multi-agent scenario, there is no explicit communication between agents during airside operations.
>
> **3. Reward shaping and credit assignment are critical in multi-agent settings.How do you ensure that individual agents receive appropriate credit for their actions to promote cooperative behavior? Are there specific reward shaping techniques employed to foster collaborative actions?**
>
> Since our approach doesn't involve multiple agents, the issues related to reward shaping and credit assignment in a multi-agent setting are not applicable. In this centralized paradigm, the actions are overseen and directed by single DRL agent, and the credit assignment is inherently tied to its decisions in managing the taxiway congestion and runway utilisation.
>
> **4. How does the method explore the joint action space and how scalable is it with increasing dimensions?**
>
> We have adopted a centralized agent paradigm to formulate the departure metering problem as it aligns with the centralized management of airside ground movements by Air Traffic Controllers (ATCOs) at airports. In other words, our problem formulation is driven by the inherent centralization in the domain, where ATCOs guide and control airside operations. Consequently, challenges related to multi-agent communication, collaboration, credit assignment, and scalability in increasing dimensions are not directly applicable to the proposed approach.

---

### Official Review · Reviewer_f91p · 2023-11-01

**Soundness:** 3 good
**Presentation:** 3 good
**Contribution:** 2 fair
**Rating:** 5
**Confidence:** 4

**Summary:**

The authors propose a novel RL-based approach for Departure Metering (DM). This is achieved by introducing domain-specific state/action representations and together with a PPO-based RL agent for this task.

Empirically, the authors show that the proposed method outperforms other methods and ablations on a simulation based on Singapore Changi Airport surface movement data.

**Strengths:**

I am not an expert in the field of air traffic control, but the domain-inspired choices in the proposed MDP representation are well motivated. The authors encode specific domain knowledge to achieve good performance.

The paper is well written, although in some parts the narration feels a bit too slow (while in others the authors gloss over a few details).

**Weaknesses:**

In my opinion, the main weaknesses of this work lie in (i) the extreme specificity of the methodology, (ii) the lack of baselines and/or experiments to validate the proposed architectural innovations, and (iii) more generally, the relevance to the broader ICLR community.

**Questions:**

(i) The extreme specificity of the methodology:
Air traffic control is a relevant problem. However, the proposed architecture seems to be extremely tailored for this one specific application. How generalizable are these methods beyond the this application? It'd be nice to see experiments on a more diverse set of problems.


(ii) The lack of baselines and/or experiments to validate the proposed innovations:
The set of baselines is extremely limited. The authors should provide additional RL-based approaches from literature or simply by implementing sensible alternative approaches to the problem.

Arguably, the major contribution of this work is the representations of MDP elements and the authors do a good job at motivating the reasoning behind their choices. It would be interesting to test how agnostic the proposed framework is to the choice of RL algorithm.

---

> ### Author Response · Authors · 2023-11-22
> **Response to reviewer questions**
>
> Thank you for your comments. Below are our responses to your questions.
>
> **1. The extreme specificity of the methodology: Air traffic control is a relevant problem. However, the proposed architecture seems to be extremely tailored for this one specific application. How generalizable are these methods beyond the this application? It'd be nice to see experiments on a more diverse set of problems.**
>
> In principle, demand-capacity imbalance problems, encountered in intelligent transportation/traffic management systems, can be tackled by adopting our proposed methodology (or parts thereof). Within Air Traffic Management (ATM) domain, for instance, the event graph idea can be potentially used for Conflict Detection and Resolution (CD-R) in en-route airspace and terminal maneuvering area. We can utilize the timed event graphs in CD-R approaches to accurately predict aircraft trajectories (represented by a sequence of way-points). Besides ATM, Markov Decision Process (MDP) based approaches have shown promise for traffic management in roads (Wei et al., 2021) and warehouses (Zong et al., 2021). Our proposed MDP framework is particularly suited to perform ramp metering where localized on-ramp delays (similar to taxi delays) must be balanced against downstream (bottleneck) freeway utilization (similar to runway utilization). In summary, our proposed method introduce several elements which can be used to solve challenging problems in transportation and logistics.
>
>
> **2. The lack of baselines and/or experiments to validate the proposed innovations: The set of baselines is extremely limited. The authors should provide additional RL-based approaches from literature or simply by implementing sensible alternative approaches to the problem.**
>
> Most DM research studies, in the extant literature, have compared DM results against non metering baselines but, have avoided comparisons with other existing methods. The reason, most likely, is that differences in assumptions, airside modelling and problem formulations make the methods incompatible on the same data set. Moreover, the lack of data (often confidential) sharing and use of different data across studies, make it particularly difficult and unfair to compare solutions. However, we understand the importance of such a benchmarking effort, for knowledge creation, for the wider academic community. Thus, we have benchmarked (performance of) our DRL algorithm against Ali et al., 2022 and Mori 2017. Ali et al. 2022 is the state-of-the-art optimization method for DM with segregated runway operations and TS is the best planning algorithm for DM under uncertainty. Besides non metering baselines, these methods were chosen as they come closest to the idea of a numerical optimization problem, on simulated airside traffic data, where expected reward is optimized with respect to the policy’s parameters. Therefore, in principle, we believe, the existing comparisons are sufficient to show our method reliability and excellent performance.
>
> **3. Arguably, the major contribution of this work is the representations of MDP elements and the authors do a good job at motivating the reasoning behind their choices. It would be interesting to test how agnostic the proposed framework is to the choice of RL algorithm.**
>
> Although not detailed in this paper due to space constraints, we conducted extensive experiments across various RL algorithms like DQN, A2C etc. We found that all RL algorithms learn and eventually converge. However, we found PPO to be the most stable and high-scoring across different traffic scenarios.
>
>
> **References**
>
> Ali, H., Pham, D. T., Alam, S., & Schultz, M. (2022). A Deep Reinforcement Learning Approach for Airport Departure Metering Under Spatial–Temporal Airside Interactions. IEEE Transactions on Intelligent Transportation Systems, 23(12), 23933-23950.
>
> Mori, R. (2017). Development of a pushback time assignment algorithm considering uncertainty. Journal of Air Transportation, 25(2), 51-60.
>
> Wei, H., Zheng, G., Gayah, V., & Li, Z. (2021). Recent advances in reinforcement learning for traffic signal control: A survey of models and evaluation. ACM SIGKDD Explorations Newsletter, 22(2), 12-18.
>
> Zong, Z., Feng, T., Xia, T., Jin, D., & Li, Y. (2021). Deep Reinforcement Learning for Demand Driven Services in Logistics and Transportation Systems: A Survey. arXiv preprint arXiv:2108.04462.

---

### Official Review · Reviewer_n3Ra · 2023-11-09

**Soundness:** 3 good
**Presentation:** 3 good
**Contribution:** 2 fair
**Rating:** 5
**Confidence:** 2

**Summary:**

This paper proposes a method based on Deep Reinforcement Learning (DRL) to address the airport takeoff control problem on mixed-mode runways. The core idea of the article includes formalizing the takeoff control problem as a Markov Decision Process (MDP), using a spatio-temporal event graph to characterize the traffic density in hotspots, and adopting a continuous deferral rate action space instead of the traditional binary open/close control method. In addition, the reward function aims to encourage high taxi speeds and runway utilization, and utilizes the Proximal Policy Optimization (PPO) algorithm to train agents. This method was evaluated on simulated traffic data from Singapore Changi Airport, and the results show that the DRL strategy reduces taxi delays, fuel consumption, and conflicts compared to baseline strategies. The article emphasizes the applicability and research characteristics of its application.

**Strengths:**

- Novel state representation using event graph captures airside congestion well

- Pushback rate action space is more practical than on/off metering

- Significant taxi delay and fuel burn reduction in experiments

- Outperforms other approaches like tabu search and baseline DRL

- Evaluated on realistic traffic scenarios from Singapore Airport

**Weaknesses:**

1) Unclear how method would transfer or scale to other airports
2) This paper focus on the application and no new algorithms is provided.
3）Lack of comparison of other optimization and planning algorithms.

**Questions:**

- How would the policy transfer to other airports with different layouts and traffic patterns?

- Could the event graph idea be used for other air traffic management tasks?

- What are other ways to set the hyperparameters instead of manual tuning?

- How would the policy perform with human controllers in the loop vs automation?

---

> ### Author Response · Authors · 2023-11-22
> **Response to reviewer questions**
>
> Thank you for your appreciation. Here are our responses to your questions:
>
> **1. How would the policy transfer to other airports with different layouts and traffic patterns?**
>
> The learnt DM policies cannot be directly applied to other airports due to different airside networks and operational constraints. However, If airport airside network information and historical surface movement data is provided, taxi routes and aircraft velocities at other airports can be simulated. Thereafter, our proposed DRL based DM framework can be easily adopted with little engineering effort which would involve clustering intersection nodes on the airside network into hotspots and applying real time trajectory prediction algorithm Tran et al. (2020) for generating a timed event graph. Thereafter, by tuning the reward coefficients ($\alpha_{1} - \alpha_{6}$), the DRL agent can be trained to perform DM, effectively balancing the airside congestion with runway utilisation, through interactions with the simulation environment.
>
>
> **2. Could the event graph idea be used for other air traffic management tasks?**
>
> Yes, the event graph idea can be potentially useful in air traffic Conflict detection and resolution (CD-R) tasks. Several studies on CD-R have employed a DRL-based approach to identify and resolve air conflicts. We can employ the timed event graph to predict aircraft trajectories, represented by a sequence of waypoints, within the DRL agent's state representation. Additionally, demand-capacity imbalance problems in air traffic management can be formulated using the event graph idea.
>
> **3. What are other ways to set the hyperparameters instead of manual tuning?**
>
> Manual hyper-parameter tuning is one of the current limitations of reinforcement learning based techniques. Although not detailed in this paper due to space constraints, we conducted extensive experiments across various neural network architectures (2, 3, 4 layers), activation functions (tanh, relu, etc.), reward coefficients ($\alpha_{1} - \alpha_{6}$) etc., aiming to determine robust hyper-parameters suitable for different traffic scenarios.
>
> **4. How would the policy perform with human controllers in the loop vs automation?**
>
> We recently performed two sets of DM experiments: first in autonomous mode and the second in human-in-the-loop (HITL) mode. In autonomous mode DM actions were implemented by the DRL agent whereas in HITL mode, the ATCOs were provided with the trained DRL based DM policy recommendations and the ATCOs were free to accept or override the recommendations. We found the DRL agent (in autonomous mode) consistently outperformed ATCOs (in HITL mode) in minimising airside delays while maintaining runway throughput. This most likely happens as, given the uncertainty at airports, it is difficult for humans to predict future traffic states and perform control actions pro-actively. Moreover, we found ATCOs—who are trained to control dedicated parts of airport airside—optimising locally for congestion whereas the DM agent would typically optimise the global airside traffic. Results from the validation experiments highlight that different ATCOs exhibit varying degrees of compliance with decision support system recommendations, indicating several factors—like personal experience, training, workload and cognitive biases—may influence their decision-making process. However, the higher degree of compliance led to a greater reduction in taxi delays.

---

### Meta-Review · Area_Chair_cwXB · 2023-12-06

**Metareview:**

This paper proposes an RL method to address the airport takeoff control problem on mixed-mode runways. This paper gives the design choices for state, action, and reward (e.g., graph-based state) for such an application and uses PPO to train agents. The method is evaluated on simulated traffic data from Singapore Changi Airport and results show the proposed method outperforms baselines.

The main weaknesses are three-fold:
1. The technical contribution is limited, mainly about how to convert takeoff control into an RL problem. No new RL algorithms are introduced.
2. The proposed method is not compared with other optimization and planning algorithms, making the proposed method not well supported.
3. It is unclear how such a method would transfer or scale to other airports.

**Justification For Why Not Higher Score:**

Refer to the weaknesses above

**Justification For Why Not Lower Score:**

N/A

---

### Decision · Program_Chairs · 2024-01-16

Reject